# OpenReview forum: "Diffusion Bridge or Flow Matching? A Unifying Framework and Comparative Analysis"
_ICLR.cc/2026/Conference — Submitted to ICLR 2026_

### Official Review · Reviewer_JEWT · 2025-10-20

**Soundness:** 3
**Presentation:** 2
**Contribution:** 2
**Rating:** 4
**Confidence:** 3

**Summary:**

The work proposed a unified formulation of Diffusion Bridge (DB) and Flow Matching (FM) models as specific cases of stochastic optimal control problems. For this, authors reformulated the dynamic optimal transport variational problem for the finding conditional flow with Wasserstein-2 transport cost as the optimal control problem and compared it with the deterministic ODE version of UniDB formulation of DB problem, which is based on the certainty equivalence principle. The main theoretical contributions of the work include: 1) proof of conditions, which lead to the degradation of FM problem to DB problem; 2) proof that the overall cost of respective optimal control problems for DB is less than for FM under specific condition of diffusion coefficient; 3) proof that the empirical McCann’s interpolation is asymptotically admits absolute continuity; 4) proof that for the quadratic cost the finite second moments ensures the validity of FM even for non-compact spaces. Experimental contributions include the quantitative analysis of the performance of DB and FM models for paired image restoration problems, including inpainting, super-resolution, deblurring, and denoising, using the same DiT architecture in the latent space of VAE and images from Celeba-HQ. Authors vary different parameters in the evaluation setup: 1) difficulty by increasing the size of masks in the inpainting problem; 2) size of the training dataset in the inpainting problem; 3) comparison in terms of training and inference time for DB and FM models. The numerical results of FID, LPIPS, SSIM, PSNR support the claims of Figure 1 about pros and cons of DB and FM models.

**Strengths:**

1) Theorem 4.2 provides a novel comparison of optimal control functionals between DB and FM models, which supports the claims of the Figure 1.
2) The proposition to use the unified architecture of DiT to have a fair comparison and the results of experiments for DB and FM models on different image restoration problems in Table 1, 2, and Figure 3, 4 correctly show different pros and cons of DB and FM stated in Figure 1. In particular, the study of quantitative results for FM and DB models depending on the size of training data proves the substantial difference between the two generative modeling paradigms.
3) Systematic profiling of training and inference time of DB and FM models in Figure 4 sums the natural efficiency of FM models due their simpler formulation.
4) The variety of 6 tested image-to-image translation problems, including image restoration and additional visual results for style transfer and semantic-to-image translation, makes the experimental study to be wide and universal enough.

**Weaknesses:**

1) The major limitation of the paper is a poor and unclear explanation of the implementation for training of DB and FM models in Section 5.1. According to the provided link for the code, one may suppose that the implementation of the DB relies on the implementation of UniDB, where for the training authors used the Algorithm 1 of UniDB in the ODE regime, while for the sampling they used the Algorithm 2 of UniDB. The implementation of FM models seems to have completely orthogonal implementation based on the code for Rectified Flow. Training and sampling pseudocodes for DB and FM models, which are similar to Algorithms 1 and 2 in UniDB, can improve the understanding of the work for the reader.
2) The comparison between DB and FM in Table 1 does not provide any information about the relation of numerical results with existing DB and FM models for image restoration problems, for example, GOUB and BoostingFM [1]. Since the evaluation protocol does not match prior works it is impossible to estimate the hardness of the considered evaluation setup without any baseline even though other DB and FM models may use different U-Net architectures.
3) The study does not provide the analysis of convergence DB and FM models with confusing information in lines 961-963. According to lines 961-963, the number of total training steps was set to 600k. At the same time, in different experiments the size of training data varied, so the training time for DB and FM models depends on the size of the training data, as shown in Table 3. There might be different criterions for the optimization stopping, which affect the results of Figure 4 and Table 3: convergence metrics/optimized objectives on validation/training data. Without these details the provided results are not clear.
4) The presentation of background for Section 4.2, Appendix A.3 and A.4 is poor, see questions 1-4 below.

References:

[1] Boosting Latent Diffusion with Flow Matching. ECCV-2024.

**Questions:**

1) Can you explain what is $T_{n}$ in the line 793? Trained image-to-image translation map on the dataset with the size $n$?
2) Is it correct that $m$ in lines 827-828 and 834-835 is a typo and there should be no $m$ in the proof?
3) In the line 830-831 is there a typo with extra comma in the formula $(1 - t)x_{i} + T_{n}(x_{i})$?
4) Can you explain in details about "By the stability above" in lines 823-824? This part in unclear.
5) Can you comment on the training-validation split in the Section 5: did all numerical results in Table 1, 2, Figure 3 represent the metrics of image-to-image translation DB and FM models on the hold out validation set? If so, what is the size of validation split and how it relates to sizes of the training data?
6) Can you comment on the stopping criterion for DB and FM models for different training data size in Table 3? How did you ensure that the convergence of those models was achieved?
7) Can you comment on the stability of training DB and FM models in Figure 3 and Table 3?
8) Since DB and FM models have the same architecture and NFE during the inference can you comment why FM models have 2-3 times faster inference?
9) Since the Figure 1 claims that DB models have longer training, but better performance compared with FM, there is question about trade-off on the quality of DB models under the same training time as FM models. Can you comment how the quality of DB models would change if you give them the same training budget as for FM models?
10) Since the Table 3 shows that the inference time of DB models is 2-3 times slower compared with FM models under the same architecture and NFE=20, can you comment on the quality degradation of DB models with the same inference time as FM models, say, when DB models use NFE=10 while FM models use NFE=20? Do we really need to run DB models with longer training and many NFE steps? Pareto curve between image quality and inference NFE for DB and FM models may improve the claims of the Figure 1.
11) Can you comment on the total number of training parameters in the DiT architecture you used for the training of DB and FM models and how it is related to other U-Net architectures used in UniDB and GOUB models?

---

> ### Author Response · Authors · 2025-11-21
> **Reply to Official Comment by Reviewer JEWT (Part 1)**
>
> We sincerely thank you for your comments and feedback.
>
> > **W1.1**: The explanation of the implementation for training of DB and FM models in Section 5.1 is poor and unclear. According to the provided link for the code, one may suppose that the implementation of the DB relies on the implementation of UniDB, where for the training authors used the Algorithm 1 of UniDB in the ODE regime, while for the sampling they used the Algorithm 2 of UniDB.
>
> A: Thanks for raising this point and here we provide a detailed explanation on the implementations. We indeed utilize UniDB [R1] as the implementation of the DB model, which is consistent with the provided code and our anaylsis. As for DB's training, we just take Algorithm 1 of UniDB **in the SDE regime, not in the ODE regime**.
>
> As for sampling, we directly adopted Euler sampling for FM. We've learned that **the Euler sampling method for FM is equivalent to diffusion with DDIMs sampler** [R2]. Therefore, to ensure a fair comparison, we adopted the first-order UniDB++ [R3] (accelerating algorithms for UniDB, similar to DDIMs accelerating for DDPMs) as the inference method of DB.
>
> [R1] Zhu et al. "UniDB: A Unified Diffusion Bridge Framework via Stochastic Optimal Control", ICML 2025.
>
> [R2] Gao et al. "Diffusion Meets Flow Matching: Two Sides of the Same Coin", ICLR 2025 blogpost.
>
> [R3] Pan et al. "UniDB++: Fast Sampling of Unified Diffusion Bridge", https://arxiv.org/pdf/2505.21528v1.
>
>
> > **W1.2**: The implementation of FM models seems to have completely orthogonal implementation based on the code for Rectified Flow.
>
> A: **Actually, our implementation is not based on Rectified Flow**. Although the corresponding file in our released code is named `rectified_flow.py`, the training and sampling process strictly follows the standard Flow Matching protocol.
>
> > **W1.3**: Training and sampling pseudocodes for DB and FM models, which are similar to Algorithms 1 and 2 in UniDB, can improve the understanding of the work for the reader.
>
> A: Yes, we have included detailed training and sampling pseudo-codes and explanations for both approaches in the revised version (Appendix D: Training and Inference Algorithms for DB and FM).
>
> > **W2**: The comparison between DB and FM in Table 1 does not provide any information about the relation of numerical results with existing DB and FM models for image restoration problems, for example, GOUB and BoostingFM [1]. Since the evaluation protocol does not match prior works it is impossible to estimate the hardness of the considered evaluation setup without any baseline even though other DB and FM models may use different U-Net architectures.
>
> A: The main target of this paper is to dissect and benchmark the intrinsic differences and performance profiles between DB and FM, instead of comparing the DB/FM to existing baselines. To ensure the fair comparison, we develop the **Same Transformer Architecture** for DB and FM. Note that **existing DB** models (like DDBMs, GOUB, UniDB) are all based on **U-Net architectures**.
>
> Here we involve the results of GOUB and UniDB (with U-Net) on 4xsuper-resolution task on CelebA-HQ dataset as a baseline for your information:
> |Super-Resolution|PSNR$\uparrow$|SSIM$\uparrow$| LPIPS$\downarrow$|FID$\downarrow$|
> | :--- | :----- | :---- | :----- | :---- |
> |GOUB|28.63|0.777|0.104|19.02|
> |UniDB|**28.70**|**0.789**|0.090|17.59|
> |FM (ours)|27.11|**0.789**|0.088|11.61|
> |DB (ours)|27.47|0.762|**0.077**|**8.50**|
>
> For super-resolution tasks, we mainly focus on the perceptual scores (LPIPS and FID) [R4]. **It is not surprise to see our DB/FM (based on Transformer) outperforms existing UniDB and GOUB (based on U-Net) significantly.** But this is not the scope of this paper. We believe conducting more experiments on BoostingFM is not necessary.
>
> [R4] Luo et al. "Image Restoration with Mean-Reverting Stochastic Differential Equations", ICML 2023.
>
> > **W3**: The study does not provide the analysis of convergence DB and FM models with confusing information in lines 961-963. According to lines 961-963, the number of total training steps was set to 600k. At the same time, in different experiments the size of training data varied, so the training time for DB and FM models depends on the size of the training data, as shown in Table 3. There might be different criterions for the optimization stopping, which affect the results of Figure 4 and Table 3: convergence metrics/optimized objectives on validation/training data. Without these details the provided results are not clear.
>
> A: Thanks for raising this concern. We emphasize that every reported comparison was conducted after both FM and DB had fully converged, i.e. we used the optimal, converged checkpoints for each method. The total 600k training steps are enough for both FM and DB.

---

> ### Author Response · Authors · 2025-11-21
> **Reply to Official Comment by Reviewer JEWT (Part 2)**
>
> We now address the remaining question.
>
> > **W4 & Q1**: Can you explain what is $T_n$ in the line 793? Trained image-to-image translation map on the dataset with the size $n$?
>
> A: Yes, you're right and we have clarified it in the revised version (marked with blue color).
>
> > **W4 & Q2-3**: Is it correct that $m$ in lines $827-828$ and $834-835$ is a typo and there should be no $m$ in the proof? In the line $830-831$ is there a typo with extra comma in the formula $(1-t) x_i+T_n\left(x_i\right)$ ?
>
> A: Thanks for raising them out and we have corrected them in the revised version (marked with blue color).
>
> > **W4 & Q4**: Can you explain in details about "By the stability above" in lines 823-824? This part in unclear.
>
> A: Thanks for point it out and we have removed the expression "By the stability above". For detailed information, please refer to the new version (marked with blue color).
>
> > **Q5**: Can you comment on the training-validation split in the Section 5: did all numerical results in Table 1, 2, Figure 3 represent the metrics of image-to-image translation DB and FM models on the hold out validation set? If so, what is the size of validation split and how it relates to sizes of the training data?
>
> A: All quantitative results in Table 1, Table 2 and Figure 3 were measured on a hold-out validation set that was never seen during training. For every training-data subset reported in the paper (500, 1000, 5000, 27000), we fixed 3000 CelebA-HQ images as the validation split, which maintains the same settings as the validation set in GOUB [R2] and UniDB [R1].
>
> [R2] Yue et al. "Image Restoration Through Generalized Ornstein-Uhlenbeck Bridge", ICML 2024.
>
> > **Q6**: Can you comment on the stopping criterion for DB and FM models for different training data size in Table 3? How did you ensure that the convergence of those models was achieved?
>
> A: In our experiments, training for both DB and FM was terminated once the model performance, especifically the FID scores, became stable. To enable continuous evaluation, we adopted a frequent checkpointing strategy so that performance could be assessed at any moment.
>
>
> > **Q7**: Can you comment on the stability of training DB and FM models in Figure 3 and Table 3?
>
> A: Figure 3 is intended solely to validate the analysis in Section 4.2: as the amount of training data shrinks, the interpolation coefficients $t$ and $1−t$ in FM gradually lose efficacy, precipitating a steep drop in performance. Table 3 summarizes the datasets, model-training costs, and inference times; it does not mention any stability issues.
>
> > **Q8**: Since DB and FM models have the same architecture and NFE during the inference can you comment why FM models have 2-3 times faster inference?
>
> A: Although two models share an identical architecture and the same NFE at inference time, their concrete implementations cause DB to run about 2× slower than FM.
>
> **Doubled channel count doubles network evaluation time.** The network input of FM is $v_{\theta}(x_t, t)$ (Line 177), whereas that of Diffusion Bridge is $\epsilon_{\theta}(x_t, x_T, t)$. Take image restoration task as an example: the DB's denoising network receives **both the current state $x_t$ and the low-quality (LQ) image $x_T$** as input, yielding **2× input channels**, whereas FM only ingests the interpolated state $x_t$. In practice, this doubles the FLOPs of every forward pass and, on modern GPUs, translates into **$\approx$ 2× wall-clock time per neural-network evaluation**.
>
> We appreciate your suggestion and have added the analysis in the revised version.
>
> > **Q9**: Since the Figure 1 claims that DB models have longer training, but better performance compared with FM, there is question about trade-off on the quality of DB models under the same training time as FM models. Can you comment how the quality of DB models would change if you give them the same training budget as for FM models?
>
> A: Yes, here we provide a comparison between DB and FM under the same training budget. Taking Inpainting (Box128) task as an example, we report the FID scores of FM and DB at the same epochs. In the early stage DB converges more slowly and delivers inferior results than FM; however, once FM has fully converged, DB continues to improve and ultimately achieves better performance.
>
> ||Epoch=40|Epoch=60|Epoch=80|Epoch=100|Epoch=150|Epoch=180|
> |:---|:-----|:----|:-----|:----|:-----|:-----|
> ||FID $\downarrow$| FID $\downarrow$ |FID $\downarrow$|FID $\downarrow$ |FID $\downarrow$|FID $\downarrow$|
> |FM (Box128)|**19.31**|**18.32**|**18.57**|18.26|18.64|18.57|
> |DB (Box128)|25.46|25.16|24.44|**15.75**|**9.32**|**7.71**|

---

> ### Author Response · Authors · 2025-11-21
> **Reply to Official Comment by Reviewer JEWT (Part 3)**
>
> We now address the remaining question.
>
> > **Q10**: Since the Table 3 shows that the inference time of DB models is 2-3 times slower compared with FM models under the same architecture and NFE=20, can you comment on the quality degradation of DB models with the same inference time as FM models, say, when DB models use NFE=10 while FM models use NFE=20? Do we really need to run DB models with longer training and many NFE steps? Pareto curve between image quality and inference NFE for DB and FM models may improve the claims of the Figure 1.
>
>
> A: Yes, here we provide a comparison between DB and FM under comparable inference time budgets. Taking Inpainting (Box128) task as an example, we compared the following two situations where FM and DB are in similar compute budgets: FM with NFE 20 (FID 17.84) vs DB with NFE 10 (**FID 9.32**), or FM with NFE 100 (FID 17.51) vs DB with NFE 50 (**FID 7.14**). It can be observed that, even when FM and DB are allotted the same **compute budget**, the performance of FM is constrained by its own model capacity, which is still falls markedly short of that achieved by DB.
>
>
> |Box128|NFE=10||NFE=20||NFE=50||NFE=100||
> |:---|:-----|:----|:-----|:----|:-----|:----|:------|:----|
> || LPIPS $\downarrow$  | FID $\downarrow$  | LPIPS $\downarrow$ | FID $\downarrow$  | LPIPS $\downarrow$ | FID  $\downarrow$ | LPIPS  $\downarrow$ | FID $\downarrow$  |
> |FM|0.111|23.01|0.106|17.84|0.099|17.40|0.101|17.51|
> |DB|**0.078**|**9.32**|**0.078**|**7.71**|**0.079**|**7.20**|**0.080**|**7.14**|
>
>
> > **Q11**: Can you comment on the total number of training parameters in the DiT architecture you used for the training of DB and FM models and how it is related to other U-Net architectures used in UniDB and GOUB models?
>
> A: Yes, the architecture employed to train both DB and FM in this paper is DiT-L/2, with approximately **485M** parameters, while the U-Net architectures used in GOUB and UniDB contains roughly **150M** parameters, which can be validated from the open-source codes.

---

### Official Review · Reviewer_D87E · 2025-10-28

**Soundness:** 2
**Presentation:** 3
**Contribution:** 2
**Rating:** 4
**Confidence:** 2

**Summary:**

This paper compares Diffusion Bridge and Flow Matching for transforming data. The authors give the first unified theory of both methods using Stochastic Optimal Control. They show that Diffusion Bridge has a lower cost than Flow Matching. They also show that Flow Matching becomes worse when using less training data.

**Strengths:**

i) The paper makes a clear connection between Flow Matching and Diffusion Bridges using a shared theoretical framework based on Stochastic Optimal Control. To the best of my knowledge, this is the first work to do so.

ii) The authors validate their theory through a wide range of experiments. They also introduce a DiT-based architecture for Diffusion Bridge.

**Weaknesses:**

i) A central claim of the paper is that lower total cost in the Diffusion Bridge framework leads to better generative performance, as stated in line 274:
   *“Lower total costs typically lead to smoother and more natural SDE/ODE trajectories, which helps to enhance the realistic visual quality of generated images and the robustness across tasks of varying difficulty.”*
   However, the theoretical justification for this claim remains unclear. While lower cost may correspond to smoother or straighter trajectories, as shown in [1, see Figure 2 and Theorem 3.5], it is not evident why this necessarily translates to improved visual quality or robustness. Given that this claim underpins much of the experimental motivation, a more detailed explanation or formal argument would significantly strengthen the paper.

ii) Several parts of the theoretical appendix are difficult to follow due to undefined or inconsistent notation. For example, in the proof of Theorem 4.2, the notation $\overline{\theta_{t:1}}$ and $\overline{\sigma_{t:1}}$ appears in line 736 without proper definition or alignment with the notation used in [2], making it hard to trace the logic. Similarly, in the proof of Remark 4.3, the term $S_{t}^{n,m}$ (line 832) is not defined, and therefore the use of limits involving $n,m \to \infty$ is not clearly justified.
   Appendix A.4 is also difficult to interpret. Although the authors claim in the main text that Flow Matching with linear schedule remains valid in non-compact latent spaces, the proof lacks a formal statement and does not make explicit use of the properties of Flow Matching models.




[1] https://arxiv.org/abs/2209.03003 Flow Straight and Fast: Learning to Generate and Transfer Data with Rectified Flow

[2] https://arxiv.org/abs/2502.05749 UniDB: A Unified Diffusion Bridge Framework via Stochastic Optimal Control

**Questions:**

i) In Section 5.5, the authors present experimental evidence that Diffusion Bridges exhibit greater stability than Flow Matching under limited training data. However, while the paper provides a theoretical explanation for the degradation in Flow Matching performance as data size decreases, no analogous theoretical justification is given for the improved robustness of Diffusion Bridges. Could the authors clarify whether this observed stability is an inherent theoretical advantage of Diffusion Bridges, or if it may be dataset-specific? In other words, is this behavior expected to hold universally, or might there be scenarios where Flow Matching is more robust?

ii) The meaning of "training time" in Section 5.5 is unclear. Figure 4 shows that Diffusion Bridges take more training time than Flow Matching. Does this mean the models were trained for different wall-clock times? If yes, that could partly explain why Diffusion Bridges perform better. Also, it is unclear why Diffusion Bridges are slower at test time under the same NFE. Since the forward pass usually takes most of the time, a detailed breakdown of runtime would help clarify this.

iii) I believe there are typos in Equations (6), (7), (8), and (9). Specifically, the constraints currently refer to $x_t$ and $dx_t$, but should likely use $x_t^u$ and $dx_t^u$ to be consistent with the controlled process notation. Could the authors confirm and correct this if necessary?

---

> ### Author Response · Authors · 2025-11-21
> **Reply to Official Comment by Reviewer D87E (Part 1)**
>
> We greatly appreciate your feedback and inquiries.
>
> > **W1**: A central claim of the paper is that lower total cost in the Diffusion Bridge framework leads to better generative performance, as stated in line 274: "Lower total costs typically lead to smoother and more natural SDE/ODE trajectories, which helps to enhance the realistic visual quality of generated images and the robustness across tasks of varying difficulty." However, the theoretical justification for this claim remains unclear. While lower cost may correspond to smoother or straighter trajectories, as shown in Reflow, it is not evident why this necessarily translates to improved visual quality or robustness. Given that this claim underpins much of the experimental motivation, a more detailed explanation or formal argument would significantly strengthen the paper.
>
> A: Yes, lower cost correspond to smoother trajectories, as shown in ReFlow, and here we provide a detailed explanation about why this necessarily translates to improved visual quality or robustness.
>
> 1. **Visual Quality.** Recall that the SOC optimization objective is the cost function $\min_{u_t}\int_0^1\tfrac12\|u_t\|_2^2 \mathrm{d}t$. UniDB [R1] states that a smaller cost functional trades for smoother, more natural diffusion paths (Proposition 4.3 [R1]). **A larger total cost implies the larger controller $\|u_t\|_2^2$**, which in turn excites oscillations along the forward trajectory and may disrupt the inherent continuity and smoothness of images [R1]. Consequently, **the state $x_t$ undergoes violent fluctuations at every time step**, destabilizing the evolution of each individual pixel and inevitably degrading the visual quality of the generated image.
> 2. **Robustness on distribution discrepancy.** We want to clarify that **the robustness here refers to the stability of the model with respect to an increasing distributional discrepancy (as the task difficulty grows)**. From the SOC viewpoint, FM employs the trivial forward ODE $\mathrm{d}x_t=u_t\mathrm{d}t$, which may be too simple to characterise the transition between arbitrary distributions. As the discrepancy between the terminal distributions grows, **the neural network becomes harder to fit the trajectories** and the optimal controller $u_t$ must compensate by increasing its magnitude, leading to highly irregular, oscillatory sample paths and noticeably degraded visual quality. However, **DB incorporates the drift term $\theta_t(x_1 - x_t)$ which helps to construct a more stable transition between two arbitrary distributions.**
>
> Thank you for raising this point, and we have added and corrected the contents in the revised version (marked with blue color).
>
> [R1] Zhu et al. "UniDB: A Unified Diffusion Bridge Framework via Stochastic Optimal Control", ICML 2025.
>
> > **W2.1**: Several parts of the theoretical appendix are difficult to follow due to undefined or inconsistent notation. In the proof of Theorem 4.2, the notation $\overline{\theta_{t: 1}}$ and $\overline{\sigma_{t: 1}}$ appears in line 736 without proper definition or alignment with the notation used in [2], making it hard to trace the logic. In the proof of Remark 4.3, the term $S_t^{n, m}$ (line 832) is not defined, and therefore the use of limits involving $n, m \rightarrow \infty$ is not clearly justified.
>
> A: Thank you for pointing them out. We have added the definitions and corrected the typos in the revised version (marked with blue color).
>
> > **W2.2**: Appendix A.4 is also difficult to interpret. Although the authors claim in the main text that Flow Matching with linear schedule remains valid in non-compact latent spaces, the proof lacks a formal statement and does not make explicit use of the properties of Flow Matching models.
>
> A: We agree that Appendix A.4 is somewhat misleading and needs rewriting: Appendix A.4 is not a formal proof; rather, it provides a certain mathematical underpinning for the validity of our experimental setup. We want to point out that although our experiments employ a VAE that maps images to a latent space which is not compact, the McCann interpolant remains perfectly valid: finite second moments of the latent codes suffice to guarantee the existence and optimality of the displacement interpolation in the quadratic-Wasserstein sense. We have made modifications to this section in the new version (marked with blue color).

---

> ### Author Response · Authors · 2025-11-21
> **Reply to Official Comment by Reviewer D87E (Part 2)**
>
> We now address the remaining question.
>
> > **Q1**: Could the authors clarify whether this observed stability is an inherent theoretical advantage of Diffusion Bridges, or if it may be dataset-specific? In other words, is this behavior expected to hold universally, or might there be scenarios where Flow Matching is more robust?
>
> A: We have clarified this observed stability is an inherent theoretical advantage of DB, not dataset-specific, **as elaborated in the Answer to W1**. Empirically, we further assessed the results on a more challenging dataset (FFHQ). As shown in the table below, the result confirms that the observed stability is an intrinsic theoretical property of DB, rather than an artifact of a particular dataset.
> ||box64||box96||box128||
> |:---|:-----|:----|:-----|:----|:-----|:----|
> ||LPIPS $\downarrow$|FID $\downarrow$|LPIPS $\downarrow$|FID $\downarrow$|LPIPS $\downarrow$|FID $\downarrow$|
> |FM|0.047|11.16|0.069|14.89|0.118|25.37|
> |DB|**0.044**|**11.11**|**0.059**|**12.02**|**0.089**|**16.72**|
>
>
> > **Q2**: The meaning of "training time" in Section 5.5 is unclear. Figure 4 shows that Diffusion Bridges take more training time than Flow Matching. Does this mean the models were trained for different wall-clock times? If yes, that could partly explain why Diffusion Bridges perform better. Also, it is unclear why Diffusion Bridges are slower at test time under the same NFE. Since the forward pass usually takes most of the time, a detailed breakdown of runtime would help clarify this.
>
> A: We emphasize that every reported comparison was conducted after both FM and DB had fully converged, i.e. we used the optimal, converged checkpoints for each method. Below we explain why DB requires longer training and inference time:
>
> **1. Why DB requires longer training time than FM?**
> **FM admits a short, direct gradient path ($x_1 - x_0$)** that bypasses intermediate numerical complications, yielding superior numerical stability. In contrast, DB must predict **a time-varying field which is intrinsically difficult to fit**, and gradients must back-propagate through a chain of compositional functions, leading to heightened numerical fragility and a slower empirical convergence rate.
>
> **2. Why DB’s inference is slower at identical NFE?**
> **Doubled channel count doubles network evaluation time.** Although two models share an identical architecture and the same NFE at inference time, their concrete implementations cause DB to run about 2× slower than FM. The network input of FM is $v_{\theta}(x_t, t)$ (Line 177), whereas that of DB is $\epsilon_{\theta}(x_t, x_T, t)$. Take image restoration task as an example: the DB's denoising network receives **both the current state $x_t$ and the low-quality (LQ) image $x_T$** as input, yielding **2× input channels**, whereas FM only ingests the interpolated state $x_t$. In practice, this doubles the FLOPs of every forward pass and, on modern GPUs, translates into **$\approx$ 2× wall-clock time per neural-network evaluation**.
>
> We appreciate your suggestion and have added the analysis in the revised version (marked with blue color).
>
> > **Q3**: There are typos in Equations (6), (7), (8), and (9).
>
> A: Yes, we have made them clarified in the revised version (marked with blue color).

---

### Official Review · Reviewer_4qbz · 2025-10-31

**Soundness:** 3
**Presentation:** 3
**Contribution:** 3
**Rating:** 6
**Confidence:** 3

**Summary:**

The paper unifies flow matching (FM) and diffusion bridges (DB) through a stochastic optimal control (SOC) formulation and shows that FM emerges as a particular case of DB. This connection enables a theoretical comparison, showing that DB typically exhibits lower SOC costs. Also, FM is shown to be less effective for small data sizes. These theoretical results are confirmed by the experiments in the controlled setup on CelebA-HQ across different tasks, data sizes and task complexity levels.

**Strengths:**

**S1.** The paper provides an interesting and novel FM formulation via SOC, showing that FM can be viewed as a special case of DB.

**S2.** Theoretical results in Theorem 4.2 and Sec 4.2 reveal two limitations of FM over DB: less stable trajectories and ineffectiveness in data-scarce regimes.

**S3.** The experiments are well-designed and conducted across various tasks of diffirent complexity levels and training data sizes. The authors also compare training and inference wall-clock times and clearly discuss the accuracy-efficiency tradeoffs.

**Weaknesses:**

**W1.** FM is only compared against the UniDB-GOU (GOUB) variant. DDBM-VP/VE also seem relevant options since both are also the special cases of UniDB. Could the authors provide the discussion on how DDBM relates to FM and compare with it?

**W2.** The experiments are limited to a single dataset (CelebA-HQ). I would suggest confirming the results on one or two more datasets (preferably more challenging).

**W3.** I recommend summarizing the training and sampling procedures for both frameworks in Appendix. This would also clarify what causes the efficiency differences.

**W4.** The results are reported only for NFE=20. Different NFEs will show the performance range for both methods and address if FM may approach DB for the same compute budget in seconds.

**Questions:**

**Q1.** Could the authors clarify why FM and DB have different training and inference times? I can see the comment in L448-449 but I still do not understand why DB is much slower for the same NFEs.

**Q2.** If I understand correctly, $\gamma$ is set to $\infty$ in all DB runs. Would $\gamma$ != $\infty$ further improve the DB advantage (according to UniDB) with no additional training/inference cost?

---

> ### Author Response · Authors · 2025-11-21
> **Reply to Official Comment by Reviewer 4qbz (Part 1)**
>
> We sincerely thank you for your constructive comments and feedback. We will provide an item-by-item response to them.
>
> > **W1**: FM is only compared against the UniDB-GOU (GOUB) variant. DDBM-VP/VE also seem relevant options since both are also the special cases of UniDB. Could the authors provide the discussion on how DDBM relates to FM and compare with it?
>
> A: In Section 6.1 of DDBMs [R1], they rigorously established that Flow Matching is recovered as the zero-noise limit of DDBMs-VE: FM is the degenerated and special case of DDBMs-VE when the stochastic volatility term vanishes. Thank you for raising this point and we have added this discussion in the revised version (marked with blue color).
>
> [R1] Zhou et al. "Denoising Diffusion Bridge Models", ICLR 2024.
>
> > **W2**: The experiments are limited to a single dataset (CelebA-HQ). The reviewer would suggest confirming the results on one or two more datasets (preferably more challenging).
>
> A: Thanks for raising this concern. We conduct further experiments on FFHQ dataset, which contains broader diversity in facial poses, ages, and ethnicities and is more challenging. As shown in the following table, FM exhibits markedly degraded stability compared to DB as the task difficulty increases (from box64 to box128), which is **consistent with our experiments in our paper**.
>
> |      | box64 |       | box96 |       | box128 |       |
> | :--- | :----- | :---- | :----- | :---- | :----- | :---- |
> |      | LPIPS $\downarrow$  | FID $\downarrow$  | LPIPS $\downarrow$ | FID $\downarrow$  | LPIPS $\downarrow$ | FID  $\downarrow$ |
> |FM|0.047|11.16|0.069|14.89|0.118|25.37|
> |DB|**0.044**|**11.11**|**0.059**|**12.02**|**0.089**|**16.72**|
>
> > **W3.1**: The reviewer recommended summarizing the training and sampling procedures for both frameworks in Appendix.
>
> A: Yes, we have added the training and sampling pseudo-codes in the revised version (Appendix D: Training and Inference Algorithms for DB and FM).
>
> > **W3.2**: This would also clarify what causes the efficiency differences. & **Q1**: Could the authors clarify why FM and DB have different training and inference times? I can see the comment in L448-449 but I still do not understand why DB is much slower for the same NFEs.
>
> A: Here we provide a more detailed explanation on different training and inference times:
>
> **1. Why DB requires longer training time than FM**
> FM admits a short, direct gradient path that bypasses intermediate numerical complications, yielding superior numerical stability. In contrast, DB must predict a time-varying field, which is intrinsically difficult to fit, and gradients must back-propagate through a chain of compositional functions, leading to slower empirical convergence rate.
>
> **2. Why DB’s inference is slower at identical NFE**
> **Doubled channel count doubles network evaluation time.** Although two models share an identical architecture and the same NFE at inference time, their concrete implementations cause DB to run about 2× slower than FM. The network input of FM is $v_{\theta}(x_t, t)$ (Line 177), whereas that of DB is $\epsilon_{\theta}(x_t, x_T, t)$. Take image restoration task as an example: the DB's denoising network receives **both the current state $x_t$ and the low-quality (LQ) image $x_T$** as input, yielding **2× input channels**, whereas FM only ingests the interpolated state $x_t$. In practice, this doubles the FLOPs of every forward pass and, on modern GPUs, translates into **$\approx$ 2× wall-clock time per neural-network evaluation**.
>
> Thanks for raising this point and we have added the discussion in the revised version.

---

> ### Author Response · Authors · 2025-11-21
> **Reply to Official Comment by Reviewer 4qbz (Part 2)**
>
> We now address the remaining question.
>
> > **W4**: The results are reported only for NFE=20. Different NFEs will show the performance range for both methods and address if FM may approach DB for the same compute budget in seconds.
>
> A: Yes, here we provided the results of different NFEs (NFE=10, 20, 50, 100) and of same compute budget in seconds.
>
> **Results of different NFEs.** Take Inpainting (Box128) as an example: as shown in the result table, FM exhibits a remarked degradation compared to DB across all the same NFEs, which demonstrates our conclusion is consistent with all NFEs.
>
> |Box128|NFE=10||NFE=20||NFE=50||NFE=100||
> |:---|:-----|:----|:-----|:----|:-----|:----|:------|:----|
> || LPIPS $\downarrow$  | FID $\downarrow$  | LPIPS $\downarrow$ | FID $\downarrow$  | LPIPS $\downarrow$ | FID  $\downarrow$ | LPIPS  $\downarrow$ | FID $\downarrow$  |
> |FM|0.111|23.01|0.106|17.84|0.099|17.40|0.101|17.51|
> |DB|**0.078**|**9.32**|**0.078**|**7.71**|**0.079**|**7.20**|**0.080**|**7.14**|
>
> **Results of same compute budget in seconds.** We compared the following two situations where FM and DB are in similar compute budgets: FM with NFE 20 (FID 17.84) vs DB with NFE 10 (**FID 9.32**), or FM with NFE 100 (FID 17.51) vs DB with NFE 50 (**FID 7.14**). It can be observed that, even when FM and DB are allotted the same **compute budget**, the performance of FM is constrained by its own model capacity, which is still falls markedly short of that achieved by DB.
>
> > **Q2**: Would $\gamma!=\infty$ further improve the DB advantage (according to UniDB) with no additional training/inference cost?
>
> A: Yes, introducing the penalty coefficient $\gamma$ could indeed improve the DB advantage without additional training/inference costs, which is the main contribution and conclusion of UniDB [R2]. $\gamma$ incurs **zero additional computational overhead** during either training or inference: it merely injects a constant offset into the existing interpolation coefficients, as explicitly demonstrated in Algorithms 1 and 2 of UniDB.
>
> [R2] Zhu et al. "UniDB: A Unified Diffusion Bridge Framework via Stochastic Optimal Control", ICML 2025.

---

> > ### Comment · Reviewer_4qbz · 2025-11-27
> >
> > I deeply appreciate the authors for their thoughtful and detailed response. I have carefully read the other reviews, the authors’ responses, and the revised paper. Most of my concerns have been well addressed, and I believe the writing is significantly improved through the added discussions, details, and refined notation. I remain positive about the submission and would like to clarify a few minor points before updating my score:
> >
> > - **2× input channels → 2× wall-clock time per neural-network evaluation.** I did not fully understand this point. Doubling the input channels is typically handled by the first model layer, which maps inputs to a fixed internal model dimension, so the cost should not scale linearly with the number of input channels. In the response to Reviewer JEWT, the authors mention that DiT-L/2 is used for both the DB and FM models. Given that, I do not see the source of a 2× FLOPs increase. Could the authors please elaborate on this?
> >
> > - **Experiments with different NFEs and FFHQ.** I thank the authors for providing these experiments. I would suggest adding them to the revision in the appendix. Please let me know if I may have missed them in the current version.

---

> > > ### Author Response · Authors · 2025-11-28
> > > **Reply to Official Comment by Reviewer 4qbz**
> > >
> > > Dear Reviewer 4qbz, thank you for your further constructive feedback.
> > >
> > > > 2× input channels → 2× wall-clock time per neural-network evaluation. The reviwer did not fully understand this point. Doubling the input channels is typically handled by the first model layer, which maps inputs to a fixed internal model dimension, so the cost should not scale linearly with the number of input channels. In the response to Reviewer JEWT, the authors mention that DiT-L/2 is used for both the DB and FM models. Given that, the reviwer does not see the source of a 2× FLOPs increase. Could the authors please elaborate on this?
> > >
> > > We kindly appreciate your insightful inquiry. In response, we have re-profiled the 20-NFE latency of both Flow Matching and Diffusion Bridge, and our measurements reveal that the CPU constitutes a non-negligible bottleneck during inference.
> > >
> > > All experiments reported in this paper were conducted on the Server (GPU: 8xH20-3e) equipped with CPUs **(Intel Xeon Platinum 8558: base frequency 800 MHz, max turbo 2.1 GHz)**. The table below summarizes the wall-clock breakdown for neural-network prediction and iterative updates in the two frameworks.
> > >
> > >
> > > | Method | Network prediction | Per-NFE Network prediction       |
> > > |--------|--------------------|---------------|
> > > | FM     | 157.73 ms          | 7.88 ms       |
> > > | DB     | 279.16 ms          | 13.95 ms      |
> > >
> > > We further benchmarked the two models on the Server (GPU: 8x4090) equipped with more powerful CPUs **(Intel Xeon Gold 6430: base 1.7 GHz, max turbo 3.4 GHz)**. The wall-clock decompositions for neural-network evaluation and iterative updates are reported below.
> > >
> > >
> > > | Method | Network prediction | Per-NFE Network prediction       |
> > > |--------|--------------------|---------------|
> > > | FM     | 182.37 ms          | 9.11 ms       |
> > > | DB     | 217.53 ms          | 10.87 ms      |
> > >
> > >
> > > And we benchmarked the two models on the Desktop (GPU: 3090) equipped with Consumer grade CPU **(Intel i7-14700K: base 3.4 GHz, max turbo 5.1 GHz)**. The wall-clock decompositions for neural-network evaluation and iterative updates are reported below.
> > >
> > > | Method | Network prediction | Per-NFE Network prediction       |
> > > |--------|--------------------|---------------|
> > > | FM     | 346.08 ms          | 17.30 ms      |
> > > | DB     | 362.29 ms          | 18.11 ms      |
> > >
> > >
> > > It can be observed that CPUs with different base frequencies have a significant impact on network prediction time. Specifically, the CPUs with larger base and max turbo frequencies would significantly reduce the gap of time overhead for FM and DB in network prediction. The reason that FM still achieved about a bit fewer network prediction time costs than DB **lies in the doubled channel**, which forces the **patch-embedding layer** to process double data, slightly raising the inference time. Sincerely thanks again for pointing out this issue and we have corrected the relevant sections (Experiment and Appendix G) in the paper (marked with blue color).
> > >
> > >
> > > > The reviewer suggest adding the experiments with different NFEs and FFHQ to the revision in the appendix.
> > >
> > > Yes, thanks for the suggestion and we have added the result tables in the Appendix F (Table 7 and Table 8).

---

### Official Review · Reviewer_rqBV · 2025-10-31

**Soundness:** 2
**Presentation:** 2
**Contribution:** 3
**Rating:** 4
**Confidence:** 2

**Summary:**

This paper provides both theoretical and experimental analysis comparing two frameworks for distribution-to-distribution sampling (beyond the standard Gaussian-to-data setting): Diffusion Bridges (DB) and Flow Matching (FM).
The contributions for this comparative analysis are two-fold:
1. **Theoretical part** The paper provides a unifying perspective by writing the DB and FM objectives as solutions of a Stochastic Optimal Control problem, with additional discussion on the choice of linear interpolation for the conditional path $x_t = (1-t)x_0 + tx_1$ in FM.
2. **Experimental part** The paper provides an experimental evaluation where both methods DB and FM are implemented with the same architecture (Latent Transformer) and tested on multiple tasks (image restoration, image translation, style transfer).

**Strengths:**

- The experimental section is detailed and shows, in a convincing manner, that DB outperforms FM on a substantial amount of the evaluated tasks. The ablation studies are relevant (including size of the train set), and a evaluation of the time cost of each method is performed.
- The code is available, and includes a novel implementation of DB using Latent Transformers, which should be valuable for the community

**Weaknesses:**

The main weakness concerns the *clarity of the theoretical part*.
Here are some comments / questions / recommendations that would make the paper easier to follow:

1. **On the SOC framework**  The paper would benefit from a clearer and more rigorous presentation. First, it would be valuable to explicitly introduce the SOC framework (in a general setting) in Section 3 (Preliminaries). Second, there are notational inconsistencies: in Equations 6 and 7, should the state be denoted $\mathbf x_t^u$ rather than $\mathbf x_t$ ?
2. **Conditional versus Total quantities** Throughout the paper, it is unclear what quantities are being analyzed. It seems that what is studied are the conditional paths $p_t(x_t | x_0, x_1)$ for DB and FM. For example, in the FM framework, the study is made on the conditional velocity field $x_1-x_0$. However, what we want to learn in the end is the true vector field $v_t (x) = \mathbb E [x_1 - x_0 | x_t]$ (even if this is performed with the conditional FM loss).
 - Could you clarify what are the motivations / interests of studying such conditional quantities ?
For example, regarding the Flow Matching discussion, the goal is typically not to have conditionally optimal paths, but rather straight trajectories when following the true velocity field (as in Rectified Flows or MiniBatch OT Flow matching).
3. **On the relation to OT** Section 4.2 would benefit from a rewriting. As it is, the main part of the section is related to discussing whether the conditional velocity field is the solution of an OT problem, which is well known (e.g. section 4.3 in Flow Matching Guide and code, Lipman et al.) and the main contribution appears to be the remark 4.3, which in this case should be highlighted.
Regarding the content of this section 4.2: does stating that Brenier theorem does not apply anymore on discrete data really imply that the linear interpolation is not well suited ? It would be interesting to articulate the practical implications of this remark and justify them in a more detailed way.

**Questions:**

See weaknesses.

---

> ### Author Response · Authors · 2025-11-21
> **Reply to Official Comment by Reviewer rqBV (Part 1)**
>
> We sincerely thank you for your comments and questions. We will provide a detailed response to these concerns.
>
> > **W1.1**: On the SOC framework The paper would benefit from a clearer and more rigorous presentation. It would be valuable to explicitly introduce the SOC framework (in a general setting) in Section 3 (Preliminaries).
>
> A: Yes, we have added Section 3.1 (Stochastic Optimal Control) to introduce the general setting of the SOC framework in Preliminaries in the revised version (marked with blue color).
>
> > **W1.2**: Second, there are notational inconsistencies: in Equations 6 and 7, should the state be denoted $\mathbf{x}_t^u$ rather than $\mathbf{x}_t$ ?
>
> A: Yes, we have corrected and made them clarified in the revised version (marked with blue color).
>
>
> > **W2.1**: Throughout the paper, it is unclear what quantities are being analyzed. It seems that what is studied are the conditional paths $p_t(x_t \mid x_0, x_1)$ for DB and FM.
>
> A: No, the conditional path is not the main topic considered in our work. Instead, we mainly investigate how the **different forward SDE/ODE processes employed by DB and FM** affect the generation performance with both theoretical and empirical analyses.
>
> We study the modeling paradigms of the forward process between DB and FM from the perspective of stochastic optimal control (SOC). Under this unfied SOC framework, we reveal that the key distinction lies in that DB incorporates the **drift term of the form $\theta_t(x_1 - x_t)$**, which is absent in FM. This different modeling substantially influences the overall cost of the SOC problems and lower costs actively guides the system toward more stable trajectories.
>
> > **W2.2**: Could the authors clarify what are the motivations / interests of studying such conditional quantities? For example, regarding the Flow Matching discussion, the goal is typically not to have conditionally optimal paths, but rather straight trajectories when following the true velocity field (as in Rectified Flows or MiniBatch OT Flow matching).
>
> A: The core motivation of this work is that although both DB and FM can achieve the transition between two arbitrary paired distributions, there remains confusion about their mutual relationship, comparative advantages, and inherent limitations. **We have clarified this motivation in Lines 11-15, 95-99, and 246-250 in our paper**.
>
> We now address the remaining question.

---

> ### Author Response · Authors · 2025-11-21
> **Reply to Official Comment by Reviewer rqBV (Part 2)**
>
> > **W3**: On the relation to OT Section 4.2 would benefit from a rewriting. As it is, the main part of the section is related to discussing whether the conditional velocity field is the solution of an OT problem, which is well known (e.g. section 4.3 in Flow Matching Guide and code, Lipman et al.) and the main contribution appears to be the remark 4.3, which in this case should be highlighted. Regarding the content of this section 4.2: does stating that Brenier theorem does not apply anymore on discrete data really imply that the linear interpolation is not well suited? It would be interesting to articulate the practical implications of this remark and justify them in a more detailed way.
>
> A: Yes, the linear interpolation does not work well when Brenier theorem is not satisfied. We have rewritten Section 4.2 **(marked with blue color)** to provide a more incisive exposition of the limitations inherent in the FM interpolation coefficients $t$ and $1-t$. Here we present some key highlights：
>
> The genuine McCann interpolant of Optimal Transport allows the velocity field to vary with $t$ while still satisfying the continuity equation, thereby minimizing the total kinetic energy. In contrast, Flow Matching’s fixed interpolation coefficients $t$ and $1-t$ effectively **freeze the velocity field**, which remains viable under small distributional discrepancy but leads to rapid performance degradation as the discrepancy widens. This failure stems from its **inability to capture the complex inter-manifold geometry**, triggering a large number of streamline intersections and **vector field conflicts**, ultimately leading to blurry learning objectives and a significant **decrease in generation quality** of the model. [R1]
>
>
> When training data are limited, the interpolation coefficients $t$ and $1-t$ lose their validity and the resulting path no longer corresponds to an optimal transport interpolation.
>
> As the training data size grows ($n \rightarrow \infty$), then $\hat{\mu}_0^{(n)} \rightharpoonup \mu_0, \hat{\mu}_1^{(n)} \rightharpoonup \mu_1$ weakly and the empirical interpolation regains absolute continuity; McCann’s interpolation is asymptotically restored, and **the performance of Flow Matching improves**. However, the coefficients remain rigidly fixed at $t$ and $1-t$, which cannot truly express the local velocity adjustment of optimal transmission. **As the discrepancy between distributions widens and the task complexity escalates, the error surges anew.**
>
> Increasing data volume alone merely mitigates, yet cannot eradicate, the systematic error of frozen vector field. **A learnable drift term $\theta_t (\boldsymbol{x}_1 - \boldsymbol{x}_t^u)$**, as incorporated by DB, is required to approximate the true Wasserstein geodesic in regimes of scarce data or complex deformation.
>
> [R1] Liu et al. "Flow Straight and Fast: Learning to Generate and Transfer Data with Rectified Flow", ICLR 2023.

---

### Author Response · Authors · 2025-12-01
**A Brief Summary of Our Rebuttal**

Dear Area Chair,

Thank you for your dedication to ensuring a timely and fair review process. We have provided comprehensive point-to-point responses to all reviewers' insightful feedback and significantly improved the manuscript through additional experiments and clarifications as suggested. Here we briefly list **key issues** raised by reviewers and describe how we have addressed them.

1. Reviewer rqBV

> It is unclear what quantities are being analyzed and what the motivations are.

The core **motivation** of this work is that although both DB and FM can achieve the transition between two arbitrary paired distributions, there remains confusion about their mutual relationship, comparative advantages, and inherent limitations.

This Reviewer misunderstood the research content of this article (the conditional path $p_t(x_t \mid x_0, x_1)$ of DB and FM). Instead, we study **how different modeling forms** of Diffusion Bridge and Flow Matching **affect the generation performation** from the perspective of Stochastic Optimal Control, and the two major shortcomings of the interpolation coefficients $t$ and $1-t$ in Flow Matching **from the Optimal Transport perspective**.

We have clarified these issues in Lines 11-15, 95-99, and 246-250 in our paper.

2. Reviewer 4qbz

> Experiment on a challenging dataset and different NFEs.

We have added more experimental results on a more challenging dataset (FFHQ) and under different NFEs (NFE=10, 20, 50, 100) in Tables 7 & 8 in the revised version.


3. Reviewer D87E

> Why does lower SOC total cost lead to improved visual quality or robustness on distribution discrepancy?

We clarified the following two parts:
- For improved visual quality, **a higher total cost** implies the larger controller, which in turn excites oscillations along the forward trajectory and can **disrupt the inherent continuity and smoothness** of images.
- For the robustness (referring to the model's stability w.r.t. an increasing distributional discrepancy as the task difficulty grows), **FM employs the trivial forward ODE** from the SOC viewpoint, which may be **too simple to characterise the transition** between arbitrary distributions. However, DB incorporates the drift term $\theta_t(x_1-x_t)$, helping to construct a more stable transition between two arbitrary distributions.

We have provided a detailed explanation in Lines 294-303 in the revised version.

4. Reviewer JEWT

> Poor and unclear explanation of the implementation for training and inference algorithms.

We have added both training and sampling pseudo-codes (Appendix D: Training and Inference Algorithms for DB and FM).

> Explain in more detail the experimental setup (such as validation set size, training time, model architecture, etc.) and the experimental results compared with other baselines.

We have provided a detailed explanation of the experimental setup, and the experimental results surpass prior works, although this is out of our paper's scope.

We also corrected some typos and added some discussions for addressing the reviewers' remaining concerns in the revised version (marked with blue color).

---
**Only one reviewer—4qbz—responded to our rebuttal before the system closed.** After reading our response, this reviewer offered quite positive feedback, as shown below:

*"I deeply **appreciate the authors for their thoughtful and detailed response**. I have **carefully read the other reviews, the authors’ responses, and the revised paper**. Most of my concerns have been **well addressed**, and I believe **the writing is significantly improved** through the added discussions, details, and refined notation. I **remain positive** about the submission and would like to clarify a few minor points before **updating my score**".*

Some **other reviewers suggested rewriting certain sections for improved clarity**. We have implemented all of these revisions in the updated manuscript. While they were unable to reply to our rebuttal directly due to the closed comment system, **Reviewer 4qbz** has **explicitly acknowledged our point-by-point responses and the subsequent revisions**.

We sincerely appreciate your time and consideration and kindly request a re-evaluation of our work.

Best regards,

The Authors

---

### Meta-Review · Area_Chair_vCAY · 2025-12-27

**Summary:**

A concern shared by most reviewers was the presentation of the paper (inconsistencies in the notation; insufficient background section; unclear motivation and focus in the main method section). Other concerns raised were limited experiments (only portrait image datasets were used; limited ablation studies, for instance missing sampling step analyses; limited baseline comparisons), as well as questions about why the diffusion bridge implementation is significantly slower than the flow matching implementation. Further, there were concerns about the theoretical justification for some claims (lower cost leads to smoother trajectories, but why does this necessarily also lead to higher quality outputs).

Further critical concerns that influenced my decision are the following: The diffusion bridge model receives both the noisy interpolated state $z_t$ and the low-quality conditioning input $z_T$ as inputs ($\boldsymbol{\epsilon}(z_t,z_T,t)$), whereas the flow matching model only receives the noisy state $z_t$ as input ($\boldsymbol{v}(z_t,t)$). Hence, the diffusion bridge receives a richer input. This is neither discussed nor studied in the paper. It would be good to train a flow matching model that also always receives $z_T$ as conditional input, thereby having a more fair comparison. Furthermore, flow matching in practice often generates new data starting from a Gaussian distribution, and conditional tasks like the ones studied in the paper are often implemented by a conditional flow matching (or diffusion) model that generates from a random Gaussian, while conditioning on the low quality input. This scenario is also not considered, but represents another critical baseline in the context of this paper.

While the concerns raised by the reviewers were partially addressed (see below), there remain open questions and critical concerns.

**Reviewer Concerns:**

The authors significantly rewrote the paper, improved the presentation, and added some additional experiments. Consequently, several concerns by the reviewers were addressed.

Meanwhile, some concerns remain. The experiments are still limited (only face image generation) and relevant baseline experiments and comparisons are missing (see above). Moreover, while the rewriting helped, some aspects are still not appropriately discussed, in particular implementation details, like the different inputs given to the model and the different inference speeds. I also think the paper could be more polished. There are many duplicated references in the bibliography, for instance.

**Reviewer Scores:**

Reviewer 4qbz gave an initial score of 6. After the initial rebuttal, the reviewer indicated willingness to potentially increase their score, potentially to an 8. All other reviewers (rqBV, D87E, JEWT) gave an initial score of 4. I think some of their concerns have been addressed in the rebuttal, so potentially at least one of these reviewers would have increased their score, to a 6. This means the paper would have landed at overall borderline scores.

---

### Decision · Program_Chairs · 2026-01-26

Reject